# Farm-to-fork changes in poultry microbiomes and resistomes in Maputo City, Mozambique

Natalie Olson,[1] Frederica Lamar,[1] Hermógenes Mucache,[2] José Fafetine,[2] Joaquim Saíde,[3] Amélia Milisse,[3] Denise R. A. Brito,[3] Kelsey J. Jesser,[4] Karen Levy,[4] Matthew C. Freeman,[1] Maya L. Nadimpalli[1]

**ABSTRACT** Increasing demand for poultry has spurred poultry production in low- and middle-income countries like Mozambique. Poultry may be an important source of foodborne, antimicrobial-resistant bacteria to consumers in settings with limited water, sanitation, and hygiene infrastructure. The Chicken Exposures and Enteric Pathogens in Children Exposed through Environmental Pathways (ChEEP ChEEP) study was conducted in Maputo City, Mozambique from 2019 to 2021 to quantify enteric pathogen exposures along the supply chain for commercial and local (i.e., scavenger) chicken breeds. Here, we performed metagenomic sequencing of total DNA from banked ChEEP ChEEP samples to characterize fecal and carcass microbiomes and resistome diversity between chicken breeds and along the supply chain. Fecal samples ($n = 26$) were collected from commercial and local chickens at production sites and markets and carcass ($n = 49$) and rinse bucket samples ($n = 26$) from markets. We conducted taxonomic profiling and identified antimicrobial resistance genes (ARGs) from metagenomic sequence data, focusing especially on potential human pathogens and "high-risk" ARGs. We estimated alpha diversity for each sample and compared by site and breed. We estimated Bray–Curtis dissimilarity between samples and examined clustering. We found that commercial and local chickens harbored distinct fecal potential pathogens and resistomes at production and market sites. Many potentially pathogenic bacteria and ARGs present in chicken fecal samples are also present on carcasses sold to consumers. Finally, commercial chicken carcasses contain high-risk ARGs that are not necessarily introduced from chicken feces. These results indicate markets are an important site of exposure to potentially pathogenic bacteria and high-risk ARGs.

**IMPORTANCE** While chicken eggs and meat are a critical protein source in low-income settings, antibiotics are routinely fed to chickens with consequences for selection of antimicrobial resistance. Evaluating how poultry gut bacterial communities, including potential human pathogens and high-risk antimicrobial resistance genes, differ from farm to market could help identify where to target interventions to minimize transmission risks to human populations. In this study in Maputo City, Mozambique, we found compositional differences between commercial and local chicken breeds at production and market sites. We also found that while all potentially pathogenic bacteria and many high-risk antimicrobial resistance genes persisted from production and market through processing, some resistance genes were detected on carcass samples only after processing, suggesting human or environmental contamination is occurring within markets. Overall, our findings indicate that open-air markets may represent a critical juncture for human exposures to pathogens and antimicrobial resistance genes from poultry and poultry products.

**KEYWORDS** poultry, metagenomics, East Africa, antibiotic resistance, markets

**Peer Reviewer** Pablo Tsukayama, Universidad Peruana Cayetano Heredia, Lima, Peru

Address correspondence to Maya L. Nadimpalli, maya.l.nadimpalli@emory.edu.

The authors declare no conflict of interest.

See the funding table on p. 14.

Demand for poultry meat in sub-Saharan Africa is projected to grow by 121% between 2005 and 2050 (1), and chicken meat and eggs provide an important source of protein. Yet, transmission of fecal pathogens from poultry represents a considerable risk to human health. In many African cities, open-air markets where livestock are purchased live and butchered on-site (2, 3) (also known as "wet" or "live" markets) comprise a significant portion of the food sector (4). These markets provide a nexus at which humans and livestock come into close contact and where livestock are processed, often without access to sufficient and clean water (3). These settings can pose a significant risk of foodborne disease transmission (5).

In addition to infection with foodborne pathogens, poultry poses additional human health risks related to the spread of antimicrobial resistance, especially in settings with poor food safety regulation and enforcement. Non-therapeutic antibiotics are often administered through chicken feed to improve growth and reduce disease, with consequences for the selection of antimicrobial resistance. A review of national monitoring programs from eight countries in the Americas, Asia, and Europe found that on average, more than 40% of *Escherichia coli* from animals were resistant to common antibiotic classes, with lower prevalences of resistance to specific classes banned for use in poultry (6). A similar review found widespread antimicrobial resistance in *E. coli* among poultry and other livestock in African countries (7). Culture-based or quantitative PCR (qPCR) methods used to quantify pathogen load in poultry do not provide insight into the diversity of antimicrobial resistance genes (ARGs) present on these food products. ARGs can pose major public health risks when mobilized (e.g., through horizontal gene transfer) by human pathogens. Metagenomic sequencing of total DNA is an increasingly popular tool to characterize entire microbial communities (i.e., the microbiome) in food and environmental samples, including the ARGs they harbor (i.e., the resistome). To date, studies using metagenomic approaches to characterize poultry microbiomes and resistomes have largely been conducted in higher-income countries (8–10), and metagenomic analyses of poultry microbiomes in low- and middle-income countries (LMICs) remain limited.

Potential differences in human health hazards presented by commercial versus local chicken breeds also remain unclear in LMIC settings. Despite growth in commercial chicken breeds worldwide, local (indigenous or native) chickens, which are often raised in households and typically scavenge for food, remain common protein sources in Asia and Africa. Local chicken breeds represented about 80% of chickens raised in LMICs in Africa and Asia as of 2006 (11), and more recent figures report approximately 70% in Rwanda and Kenya as of 2015 (12). In Maputo City, Mozambique, while 45% of surveyed households purchased and 62% consumed broiler chicken meat in the past week, only 1% purchased and 3% consumed local chicken meat in the past week (13). In poultry farms and domestic chickens in Ecuador, significantly higher levels of antimicrobial resistance were measured among commercially raised chickens than among local chickens (14). Local chickens are typically locally adapted and resistant to regionally circulating disease (15), but immunity nevertheless remains a chief concern among local chicken farmers in Kenya (16). While broiler and layer chicken breeds have been bred to optimize feed efficiency and growth, this may come at the cost of immune traits (17). For example, greater resistance to *Salmonella typhimurium* infection among local chicken breeds than among broiler chicken breeds (18) suggests that immune regulation may be compromised in broiler and layer chickens as compared with local breeds, necessitating prophylactic antibiotic use in commercial chicken production. It remains unclear whether commercial and local chickens bear different microbiomes or resistomes and therefore may represent different hazards to human health.

African countries bear the greatest burden of foodborne illness; the World Health Organization region, including Mozambique, reports a foodborne illness burden of 1,200 disability-adjusted life-years (DALYs) per 100,000 people (19). Poultry, in particular, appears to pose acute risks; poultry represents the greatest source of foodborne illness compared with other meat and vegetable products, representing an annual

burden of 65,000 and 42,600 DALYs, respectively (20). Our recent study in Maputo City, Mozambique, reported significantly higher concentrations of *Salmonella enterica* and *Campylobacter* spp. DNA in poultry carcass meat at live bird markets than at farms, suggesting amplification along the value chain (21). While absolute quantification of specific pathogens in chickens can help inform risk assessment, the diversity of pathogens that may be contaminating these products remains unknown.

Here, we employed metagenomic sequencing to characterize potentially pathogenic bacteria and ARGs harbored by chickens and chicken products at production and market sites in Maputo City, Mozambique. This study was nested within the Chicken Exposures and Enteric Pathogens in Children Exposed through Environmental Pathways (ChEEP ChEEP) study, which aimed to identify pathways of exposure to enteric pathogens among children in Maputo City, Mozambique (22). A prior ChEEP ChEEP study revealed increasing contamination of chickens with enteropathogenic *Campylobacter* spp. throughout the value chain from farm to market, resulting in 100% contamination of carcasses sold to consumers (23). Given this ubiquitous and escalating *Campylobacter* spp. contamination along the poultry value chain, we sought to characterize the full microbiomes and resistomes of these samples to more comprehensively characterize potential risks.

## MATERIALS AND METHODS

The ChEEP ChEEP study was conducted between 2019 and 2021 in Maputo City, Mozambique. Pooled fecal samples (*n* = 136) were collected from broiler, layer, and local chickens at farms, households, and markets. Commercial and local chickens are sold at the same markets in Maputo City; fecal samples were collected from cages after arriving at markets. Sample collection and processing methods are described in detail in prior ChEEP ChEEP study publications (23, 24). Sterile spatulas were used to collect pooled fecal samples from broiler and layer housing at small-scale farms, yards and holding cages at households, and shared holding cages at markets; samples and a daily field blank were transported to the laboratory on ice and processed the same day. For each chicken breed and site, four fecal samples were pooled by equal weights. DNA was extracted from ~1 g of each pooled fecal sample using Qiagen PowerBead DNA extraction tubes (23).

Rinses of chicken carcasses, hereafter referred to as "carcass samples" (*n* = 75), were collected from broiler, layer, and local chickens at markets between 2019 and 2021 (23). Raw carcasses were placed in Whirl-Pak poultry rinse bags at the market, transported to the laboratory on ice, and processed the same day. Individual carcasses were rinsed with 400 mL 0.1% buffered peptone water in Stomacher bags by shaking and moving in an arc motion for 1 minute. One hundred milliliters of the resulting solution was aliquoted into two 50-mL conical tubes and centrifuged. Resultant pellets were resuspended in 1 mL 1× phosphate buffered saline, and then 125 µL of each aliquot (250 µL total for each carcass) was transferred to PowerBead DNA extraction tubes. Water collected from rinse buckets used by vendors for processing broiler chickens was additionally collected from three markets between 2021 and 2022 (24), hereafter referred to as "rinse bucket samples." We refer to carcass and rinse bucket samples collectively as "processing samples."

To preserve the integrity of genetic material in our samples, Qiagen DNEasy PowerSoil Kit extraction buffer "Solution C1" was added to each sample. Six freeze-thaw cycles were completed to break open *Cryptosporidium* spp. oocysts. DNA was extracted from all samples using DNEasy PowerSoil Kit and stored at −80°C. Invitrogen Qubit fluorometry and NanoDrop were used to measure the DNA concentrations in each extract, and all samples used for sequencing were >10 ng/µL.

Extracted DNA from a subset of these samples (*N* = 101) was available in sufficient quantity (>10 ng/µL threshold concentration) for metagenomic sequencing (Table S2). Available samples included fecal, carcass, and rinse bucket samples collected from

broiler, layer, and local chickens at three markets, four farms, and seven households (Fig. 1). For the purposes of this study, samples from broilers and layers were collectively considered "commercial" chicken samples.

## Data analysis

### *Metagenomic sequencing and quality control*

NEBNext Ultra II DNA Library Prep Kit was used to for library preparation. These samples underwent short-read paired-end 150-bp sequencing by Novogene (Sacramento, CA, USA), using the Illumina NovaSeq 6000 PE150 platform. We performed all bioinformatics analyses using tools installed on the Rollins School of Public Health High Performance Computing Cluster (25). We removed adapters and low-quality reads from raw read files using BBDuk version 38.9 (26). We used Bowtie2 version 2.5.1 (27) with National Institutes of Health Genome Reference Consortium reference genomes to align and remove host and eukaryotic contaminant DNA sequences representing chicken (28), mallard duck (29), tufted duck (30), pheasant (31), marmot (32), quail (33), swan (34), and maize (35).

### *Taxonomic classification*

We used a reduced 500-GB National Center for Biotechnology Information (NCBI) Reference Sequence taxonomic database (36) to conduct taxonomic profiling using Kraken2 version 2.1.2 with default k-mer length of 35 (37) and used Bracken version 2.7.0 (Bayesian Re-estimation of Abundance after Classification with Kraken) (38) using default settings to estimate relative abundance of classified bacteria. Because chicken fecal, carcass, and rinse bucket samples contain many bacterial species that are not relevant for human health, we subset microbial species to established human pathogens as determined by Bartlett et al. (39) for the purposes of analysis. These will be reported as "potential pathogens" or "potentially pathogenic bacteria." We also report analyses for all bacterial species, regardless of pathogen status.

### *ARG profiling*

We used k-mer alignment (KMA) version 1.4.9 (40) to align trimmed and host-decontaminated reads to the ResFinder database (downloaded June 2, 2023) (41) using a threshold of 95% identity and 90% coverage (40). To distinguish ARGs that pose an elevated risk to humans, we identified those that scored in the highest quartile "health risk to humans, defined as the risk that ARGs will confound the clinical treatment for pathogens" as determined by Zhang et al. (42) based on "human accessibility, mobility, pathogenicity and clinical availability." We hereafter refer to these ARGs as high-risk ARGs (HR-ARGs); a list of HR-ARGs is available in the supplemental material. We report analyses for all ARGs as "ARGs."

To estimate the total number of unique organisms present in our data, we estimated genome equivalents for each sample metagenome using Microbe Census version 1.1.1 (43). To quantify ARGs in the context of total bacterial genomes, we calculated fragments per genome equivalent (FPGE) for each ARG. The resulting value was then log(10) transformed.

$$\text{FPGE} = \frac{\text{sequencing depth}}{\frac{\text{template length}}{1,000} * \frac{\text{genome equivalents}}{1,000,000}}$$

### *Statistical analysis*

For all fecal and carcass samples, we estimated alpha (within-sample) diversity measures including species richness (total unique species in each sample) using Breakaway (44) and species evenness (inverse Simpson diversity index) using DivNet (45). We used Wilcoxon signed-rank test statistics to compare alpha diversity measures between commercial versus local chickens at each site (i.e., at the site of production versus at

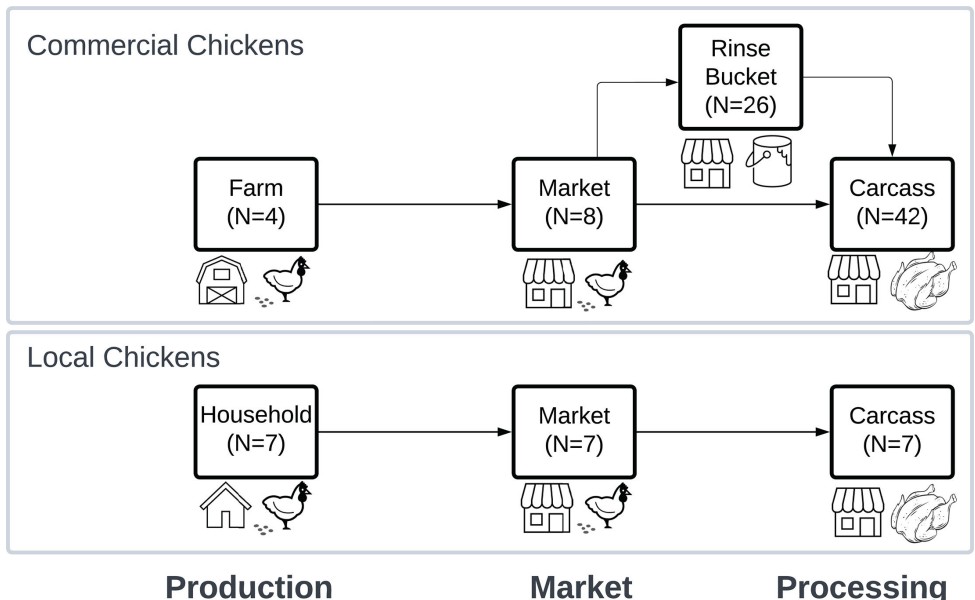

**FIG 1** Diagram of samples collected along the poultry value chain. Samples were collected from commercial (top) and local (bottom) chickens. Comparisons were made between breeds for fecal samples at production and market sites and carcass processing samples. Samples within each breed were compared between farm and market and market and processing.

markets) and between sites for each breed. We estimated beta (between-sample) diversity by calculating Bray–Curtis distances between samples. We estimated these measures using all classified bacteria in each sample and separately using only the subset classified as established human pathogens. We visualized compositional patterns using non-metric multidimensional scaling (NMDS) ordination of Bray–Curtis distances (46). We tested differences in community composition between groups using permutational multivariate analysis of variance (PERMANOVA). We modeled differences in the relative abundance of total bacterial genera and species, potentially pathogenic bacteria and species, and HR-ARGs between samples using beta-binomial regression with Corncob (47). Differentially abundant taxa were identified in Corncob using Wald tests and after adjusting for false discovery rate (FDR). Differentially abundant ARGs were identified in Corncob using the "differentialtest()" function to test the hypothesis that the ratio of the mean abundance of HR-ARGs in the referent group relative to the comparator group was larger than typical among observed ARGs.

## RESULTS

### Production and market: fecal samples

We identified 22,986 bacterial species in chicken fecal samples, of which 10 (<1%) were uniquely identified in commercial chickens and 17 (<1%) were uniquely identified in local chickens. Bacterial reads comprised 49% of total reads in fecal samples (Table 1), on average. Among unique bacterial species detected in chicken fecal samples, 118 (0.5%) were potential pathogens, comprising 13% of total bacterial reads. All detected potential pathogens were observed in both commercial and local chicken feces. We detected 164 unique ARGs in fecal samples, of which 60 (37%) were uniquely identified among commercial chickens and 28 (17%) were uniquely identified among local chickens. Among all ARGs identified in fecal samples, 36 (22%) were identified as HR-ARGs. Of these HR-ARGs, eight (22%) were uniquely detected in commercial chickens, and five (14%) were uniquely detected in local chickens. Compared with fecal samples, we identified substantially fewer bacterial reads in chicken processing samples, including carcass (16% of total reads among commercial and 6% of total reads among local

chickens, on average) and rinse bucket samples (27% on average), likely reflecting high degrees of contamination by host DNA from chickens, humans, and other sources.

## Commercial and local chicken potential pathogens and resistomes are compositionally distinct from each other at production sites

Potentially pathogenic bacteria ($P$ = 0.003) and ARGs ($P$ = 0.018) differed between commercial and local chickens at production sites, that is, farms for commercial chickens and households for local chickens, as indicated by PERMANOVA testing of Bray–Curtis distances after ordination with NMDS (Fig. 2A). Bacterial communities in general ($P$ = 0.103) and HR-ARGs ($P$ = 0.070) were not significantly different between breeds. At production sites, potentially pathogenic *Listeria* spp. ($P_{adj}$ = 0.024), *Campylobacter* spp. ($P_{adj}$ = 0.041), *Salmonella* spp. ($P_{adj}$ < 0.001), *Klebsiella* spp. ($P_{adj}$ < 0.001), *Yersinia* spp. ($P_{adj}$ = 0.022), *Pseudomonas* spp. ($P_{adj}$ = 0.018), *Vibrio* spp. ($P_{adj}$ = 0.032), and *Leptospira* spp. ($P_{adj}$ = 0.041) were less abundant among local than commercial chickens after adjusting for FDR (Fig. 2B). Potentially pathogenic *Escherichia* spp. ($P_{adj}$ < 0.001) was the only genus more abundant among local than commercial chickens at production sites. Among 129 unique ARGs detected in commercial and local chicken fecal samples at the site of production, 51 (40%) were detected in both breeds, 60 (46%) only in commercial chickens, and 18 (14%) only in local chickens. The most abundant HR-ARGs among both breeds were *tetQ*, *tetW*, *ant6-Ia*, *ermF*, and *mefA*. Multiple high-risk tetracycline resistance genes, including *tet40* ($P_{adj}$ = 0.013) and *tetX* ($P_{adj}$ = 0.013), were significantly more abundant in commercial chicken feces than local chicken feces (Fig. 2C and D).

## Commercial and local chicken fecal microbiomes and resistomes are compositionally distinct from each other at market sites

Commercial and local chickens are sold at the same markets in Maputo City; fecal samples were collected from cages after arriving at markets. Despite sharing the same market environment, we observed that commercial and local chickens harbored distinct bacterial communities ($P$ = 0.0001) and ARGs ($P$ = 0.0001) at market as indicated by PERMANOVA testing of scaled Bray–Curtis distances (Fig. 3A). Both breeds harbored distinct communities of potentially pathogenic bacteria ($P$ = 0.003) and HR-ARGs ($P$ = 0.001). Potentially pathogenic taxa that were differentially abundant between breeds at the site of production, for example, *Listeria* spp., *Campylobacter* spp., and *Salmonella* spp., were not differentially abundant at market. However, potentially pathogenic species of the genus *Enterococcus* ($P_{adj}$ = 0.002) were significantly less abundant among household than commercial chickens at market after adjusting for FDR (Fig. 3B). Of all the HR-ARGs examined, only the e*rm(F)* gene ($P_{adj}$ = 0.004), which confers resistance to erythromycin antibiotics, was more abundant among commercial than local chicken fecal samples (Fig. 3C and D) at market; previous differences in tetracycline resistance gene abundance that we observed between breeds at production sites were not apparent at market. Among 122 unique ARGs detected in market fecal samples, 55 (45%) were detected in both breeds at market sites, 41 (34%) only among commercial chickens, and 26 (21%) only among local chickens.

Commercial chickens sampled at markets harbored a greater number of bacterial species ($P$ = 0.002) and a greater number of potentially pathogenic species ($P$ = 0.006) than commercial chickens sampled at the farm level, though we observed no differences in species richness, diversity, or taxa abundance among local chicken feces sampled at households versus at markets. Potentially pathogenic *Listeria* spp. ($P_{adj}$ = 0.035), *Salmonella* spp. ($P_{adj}$ < 0.001), *Klebsiella* spp. ($P_{adj}$ < 0.001), *Enterobacter* spp. ($P_{adj}$ < 0.001), *Pseudomonas* spp. ($P_{adj}$ ≤ 0.001), *Vibrio* spp. ($P_{adj}$ ≤ 0.001), and *Treponema* spp. ($P_{adj}$ ≤ 0.001) were relatively less abundant among commercial chickens sampled at markets versus at farms after adjusting for FDR. Commercial chickens also harbored distinct communities of ARGs at markets versus at farms ($P$ = 0.023). Among 136 unique ARGs detected in commercial chicken fecal samples, 71 (52%) were detected among

**TABLE 1** Characteristics of 101 metagenomes of poultry feces, carcass, and rinse bucket samples from along the poultry value chain in Maputo, Mozambique, 2019–2022

| | | Production (fecal) | | | Market (fecal) | | | Processing (carcass) | | |
|---|---|---|---|---|---|---|---|---|---|---|
| | | Commercial (n = 8) | Household (n = 7) | P-value | Commercial (n = 4) | Household (n = 7) | P-value | Commercial (n = 42) | Household (n = 4) | P-value |
| Bacteria | Mean (SD) total reads | 48.7 (25.7) | 34.1 (18.3) | 0.073 | 28.5 (6.7) | 22.1 (12.4) | 0.955 | 4.4 (2.5) | 3.0 (0.63) | 0.955 |
| | Mean (SD) reads (millions) | 18.8 (6.5) | 18.4 (7.5) | 0.927 | 15.2 (5.4) | 9.9 (7.8) | 0.072 | 1.1 (2.4) | 0.18 (0.09) | 0.072 |
| | Proportion of total reads | 0.41 (0.09) | 0.56 (0.12) | 0.042 | 0.52 (0.08) | 0.44 (0.23) | 0.121 | 0.16 (0.21) | 0.06 (0.04) | 0.121 |
| | Mean (SD) estimated richness | 22,612 (239) | 22,441 (324) | 0.527 | 22,261 (183) | 22,204 (267) | 0.613 | 25,580 (28,834) | 24,130 (13,739) | 0.727 |
| | Inverse Simpson Diversity Index | 61.4 (33.1) | 27.8 (12.9) | 0.042 | 44.0 (23.4) | 39.7 (39.2) | 0.397 | 14.2 (9.2) | 13.1 (8.5) | 0.9 |
| Pathogens | Mean (SD) reads (millions) | 0.96 (0.55) | 3.15 (4.91) | 0.412 | 1.0 (0.68) | 3.2 (6.6) | 0.613 | 0.06 (0.11) | 0.01 (0.01) | 0.613 |
| | Mean (SD) proportion of bacterial reads | 0.05 (.03) | 0.18 (0.28) | 0.412 | 0.07 (0.04) | 0.21 (0.25) | 0.189 | 0.07 (0.07) | 0.08 (0.05) | 0.189 |
| | Mean (SD) estimated richness | 1,263 (19) | 1,268 (16) | 0.667 | 1,238 (27) | 1,237 (67) | 0.667 | 1,235 (917) | 977 (398) | 0.566 |
| | Inverse Simpson diversity index | 5.7 (2.8) | 2.6 (1.0) | 0.024 | 4.6 (2.2) | 3.0 (1.6) | 0.189 | 4.7 (3.2) | 2.7 (0.6) | 0.078 |

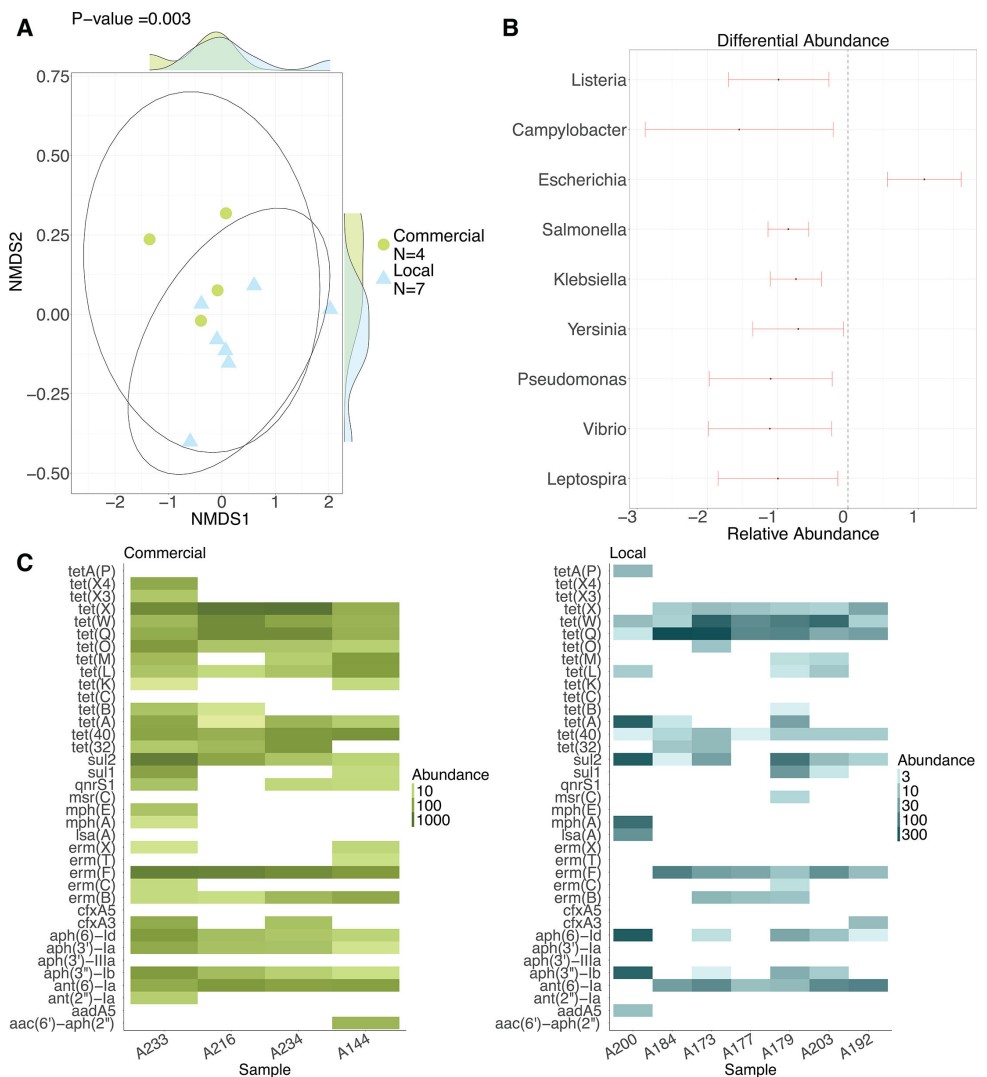

**FIG 2** Commercial and local chicken fecal microbiomes and resistomes are compositionally distinct at production sites. (A) NMDS plot of Bray–Curtis distances for pathogen composition. (B) Differences in relative abundance of pathogens among local versus commercial (reference) chickens, modeled using beta-binomial regression. (C) Heat map of HR-ARG abundances among commercial and local chickens.

both farm and market samples, 40 (30%) among only farm samples, and 25 (18%) among only market samples.

## Processing: carcass and rinse bucket samples

We detected 22,851 bacterial species in carcass samples, of which 4,700 (21%) were uniquely identified in commercial chicken carcass samples and 68 (<1%) were uniquely identified in local chicken carcass samples. Among bacterial species, 118 (0.5%) were identified as potentially pathogenic bacteria, of which 13 (11%) were uniquely detected in commercial chicken carcass samples. All potential pathogens identified in local chicken carcass samples were also observed among commercial chicken samples. The most abundant bacterial genera across all processing samples were *Aeromonas* spp., *Acinetobacter* spp., *Chlamydia* spp., and *Escherichia* spp. Compared to chicken fecal samples, a substantially lower percentage of bacterial species detected in chicken processing samples was potential human pathogens (7%). We detected 141 unique ARGs in processing samples, of which 140 (99%) were uniquely identified in commercial chicken processing samples. Only one ARG was detected among local chicken

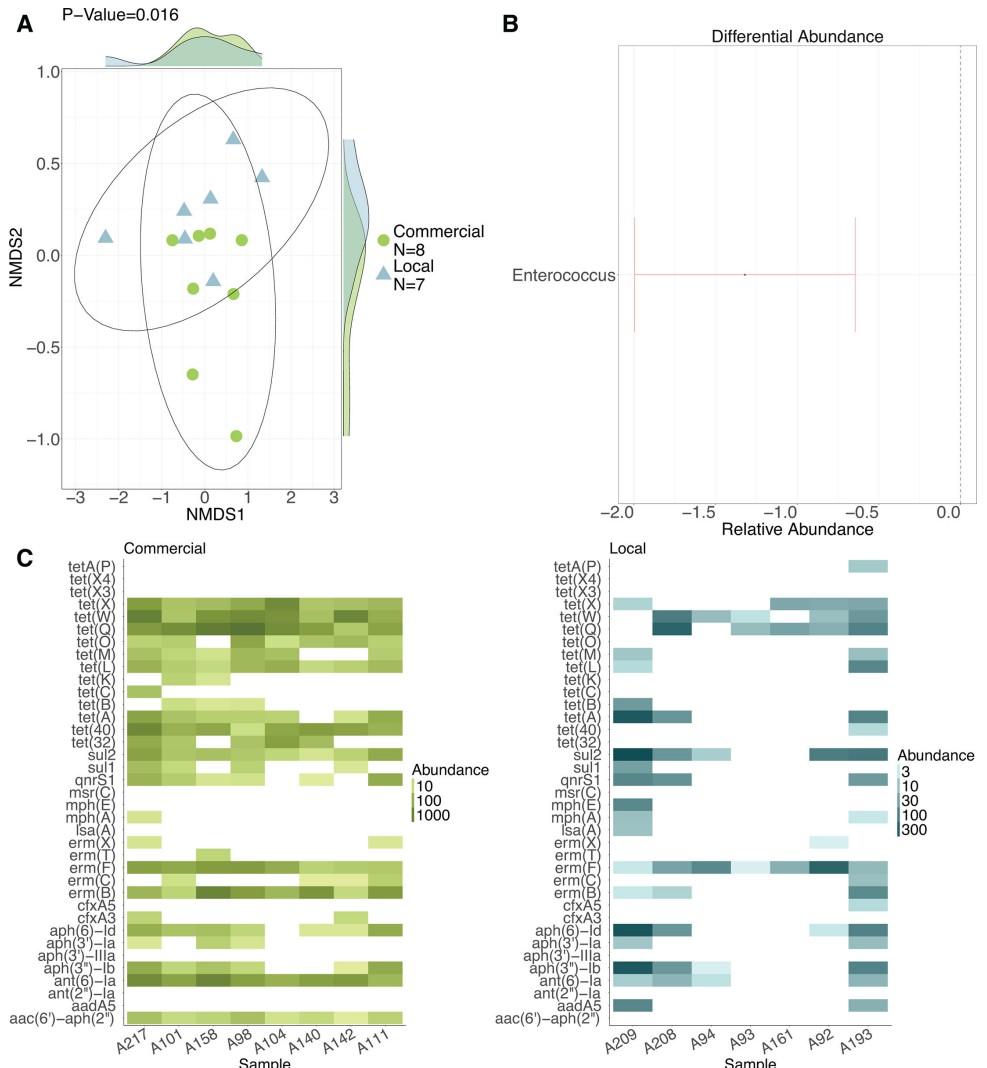

**FIG 3** Commercial and local chicken fecal microbiomes and resistomes are compositionally distinct at market sites. (A) NMDS plot of Bray–Curtis distances for pathogen composition. (B) Differences in relative abundance of pathogens among local versus commercial (reference) chickens, modeled using beta-binomial regression. (C) Heat map of high-risk HR-ARG abundances among commercial and local chickens.

processing samples (*tetE*), although this ARG was also detected in commercial chicken processing samples. HR-ARGs were only detected in commercial chicken processing samples; 26 unique HR-ARGs were detected.

## Many, but not all, potentially pathogenic bacteria and ARGs present in chicken fecal samples are also present in carcass samples sold to consumers

Among both commercial and local chickens, all potentially pathogenic bacteria detected in fecal samples from production and market sites were also detected in carcass samples. The most abundant potential pathogens on chicken carcasses included *Aeromonas* spp. (56% of potential pathogen reads), *Acinetobacter* spp. (13%), *Chlamydia* spp. (6%), and *Escherichia* spp. (3%). Commercial and local chicken carcasses harbored significantly different communities of potential pathogens (*P* = 0.04), mirroring patterns observed in chicken feces. Similarly, HR-ARGs detected in commercial chicken fecal samples were also detected on commercial chicken carcasses, with *Tet39* (28% of reads), *srmB* (19%), and *sul2* (18%) being the most abundant HR-ARGs on commercial chicken carcasses that

likely originated from chicken feces. Unusually, while HR-ARGs were detected in local chicken feces, we did not detect any HR-ARGs among local chicken carcasses (Fig. 4A).

## Commercial chicken carcasses contain HR-ARGs that are not necessarily introduced from chicken feces

Among 166 unique ARGs detected among commercial chicken fecal and carcass samples at market sites, 70 (42%) were detected among carcass samples only. Among the 70 ARGs that were uniquely detected in commercial chicken carcass samples that were not detected in the feces of commercial chickens at market, the most abundant were *qnrS2* (7.04 log10 FPGE, detected in 21 of 42 samples), which confers resistance to quinolone antibiotics; a*mpS* (6.75 log10 FPGE, in 13 samples), which confers resistance to ampicillin; *crpP* (6.69 log10 FPGE in one sample), which can confer resistance against ciprofloxacin (48); and *tetE* (6.68 log10 FPGE in 26 samples) and *tetS* (6.13 log10 FPGE in 16 samples), which both confer resistance to tetracycline antibiotics. A total of 45 (64%) of the 70 detected ARGs were found in rinse bucket and carcass samples, but not fecal samples, suggesting that rinse bucket water could be a potential source of these ARGs to chicken carcasses (Fig. 4B). Among 30 HR-ARGs detected among commercial chicken fecal and carcass samples at market sites, 21 (70%) were detected among both fecal and carcass samples, six (20%) among fecal samples only, and three (10%) among carcass samples only (Fig. 4B). Three (10%) HR-ARGs that were uniquely detected in carcass but not fecal samples included *ant(2)-Ia aph(3)-IIIa*, and *mphE*. Of these, only *aph(3)-IIIa* was uniquely detected in carcass but not rinse bucket samples. HR-ARGs *aadA5* and *ermC* were detected exclusively in rinse bucket and not in carcass samples, and 23 other HR-ARGs were detected in both carcass and rinse bucket samples. There was no significant difference in overall composition of ARGs between rinse bucket and carcass samples (*P* = 0.63).

## DISCUSSION

We employed metagenomic shotgun sequencing to characterize and compare poultry microbiomes and resistomes at multiple points along the supply chain in Maputo City, Mozambique. We took this approach to broadly characterize potential pathogen and ARG exposures at multiple points along the supply chain to understand and compare potential hazards to human health from poultry and poultry products.

While targeted amplification of enteropathogenic *Campylobacter* spp. and *Salmonella* spp. revealed increasing contamination of chicken fecal and carcass samples along the value chain in the ChEEP ChEEP study (23), this follow-up analysis reveals a wider array of potential pathogens and ARGs in a subset of the same fecal and carcass samples. Potential human pathogens and HR-ARGs persisted along the poultry value chain and contaminated slaughtered chicken carcasses purchased by consumers. While most ARGs appeared to originate in chicken feces, many did not, suggesting alternate sources of contamination in the market setting. Rinse bucket water is a likely source of additional contamination, especially given that 100% of chickens tested positive via PCR for *Campylobacter coli/jejuni* and *E. coli* after being rinsed, despite fewer chickens testing positive for these species prior to rinsing (24). Additionally, other environmental sources, such as vendors, surfaces, and pests, could explain the greater diversity of potential pathogens and ARGs observed on carcasses. A transmission modeling study parameterized using ChEEP ChEEP data estimated that 90% reduction in *Campylobacter* and *Salmonella* spp. contamination at these markets would result in meaningful reductions in infection with *Campylobacter* and *Salmonella* spp. among consumers (49). Our findings align with this and similar studies suggesting markets as a likely site of important exposures to pathogens and ARGs from poultry and poultry products (3, 49).

We observed differences in potentially pathogenic bacteria and ARG abundances between commercial and local chickens that persisted along the value chain. At production sites, local chicken fecal samples carried significantly lower relative abundance of potentially pathogenic bacterial genera including *Listeria*, *Campylobacter*,

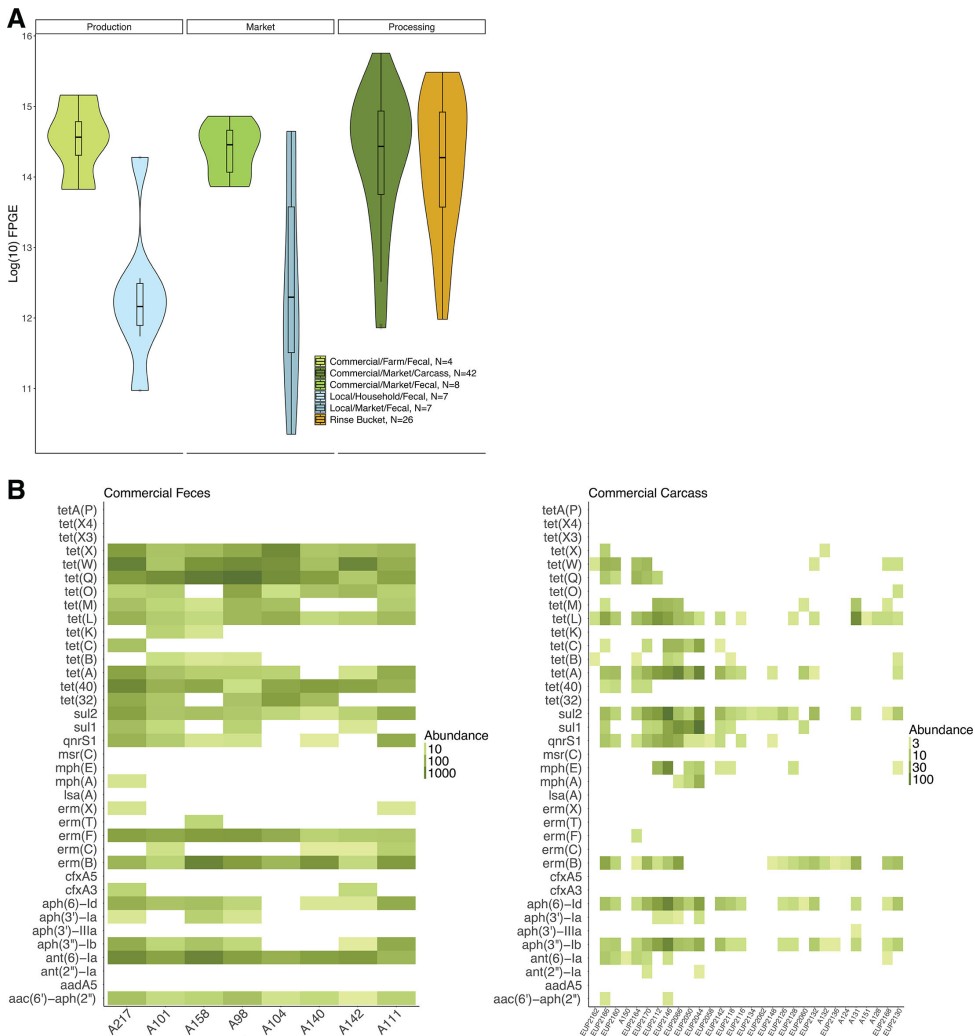

**FIG 4** Consumers are exposed to clinically important HR-ARGs via chicken carcasses that are not necessarily introduced from chicken feces. (A) Violin plot depicting HR-ARG abundance measured in FPGE of local and commercial chicken samples at production, market, and processing. HR-ARGs are abundant at multiple points in the supply chain, especially among commercial chickens. Note that local carcass samples are missing from this plot as no HR-ARGs were detected in these samples. (B) Heat map of HR-ARG abundances among commercial fecal and carcass samples. HR-ARGs *ant(2)-Ia*, *aph(3)-IIIa*, and *mphE* were identified in commercial carcass but not market fecal samples.

*Escherichia*, *Salmonella*, *Klebsiella*, *Yersinia*, *Pseudomonas*, *Vibrio*, and *Leptospira* than commercial chickens. Greater potential pathogen burden among commercial chickens at production sites may result from age differences between groups, with commercial chickens sampled at farms likely younger than local chickens sampled at households. Chicken microbiome abundance and diversity reach a plateau at 7 days of age, but compositional changes continue to occur throughout their development (50). Further research is needed to parse whether the differences we observed in potential pathogen load at the site of production were due to differences in chicken age or breed. We also observed that commercial chicken fecal microbiomes and resistomes, but not those of local chickens, differed between production and market sites. This difference could be a result of age. According to studies in eastern and southern Africa, although commercial and local chickens brought to market have reached maturity, commercial chickens are sold at markets at an average age of 60 days old (51) compared with local chickens at an average age of 24 weeks (52). Alternatively, it is also possible that stress induced by commercial transportation factors (53), including heat (54) and crowding (55), resulted in

different microbiome profiles between farm and market commercial chicken fecal samples. Importantly, while commercial chicken fecal microbiomes and resistomes differed between farm and market, they continued to remain compositionally distinct from local chickens. We hypothesized that market environments would provide opportunities for pathogen and antimicrobial-resistant bacterial transmission among poultry of distinct breeds and origins. However, we found no evidence that commercial and local chicken microbiomes or resistomes were more similar at market than production sites; rather, each continued to pose distinct hazards to consumers.

We observed that bacteria and ARGs harbored by chickens at farms and markets may persist through processing and therefore could pose a threat to the health of consumers. The most abundant potentially pathogenic species detected in carcass samples included *E. coli*, *Klebsiella pneumoniae*, *S. enterica*, and *Shigella flexneri*, as well as the opportunistic pathogen *Enterobacter hormaechei*, which is often seen in healthcare-associated infections (56). Detection of these potentially pathogenic species is notable because consumption of contaminated poultry and egg products is commonly linked with human salmonellosis cases. In the United States, over 70% of human salmonellosis (*S. enterica*) cases are associated with poultry and egg consumption (57). *K. pneumoniae* are opportunistic pathogens known to commonly harbor and even "traffic" ARGs from environmental bacteria to human pathogens, for example, extended-spectrum beta-lactamase genes (58), and are therefore considered a significant threat to public health (59–61). *Tet39*, which confers resistance against tetracycline (62), was the most abundant HR-ARG and was detected only in commercial chicken carcass samples. Tetracycline resistance was reported in 62% of 120 *Staphylococcus aureus* isolates tested in a similar "farm-to-fork" study of poultry and poultry products tested in South Africa (63) and in 100% of 24 *Campylobacter* spp. isolates from broiler chicken carcasses sampled at markets and slaughterhouses in Maputo City, Mozambique (64).

We observed some HR-ARGs only in carcass samples, but not fecal samples. Several HR-ARGs, including *ant(2)-Ia*, *aph(3)-IIIa*, and *mphE,* were detected only in processed carcass samples and not at previous points along the value chain. The ARGs a*nt(2)-Ia* (65) and *aph(3)-IIIa* (66) confer resistance against aminoglycosides (including gentamicin, amikacin, tobramycin, neomycin, and streptomycin) while *mphE* (62) confers resistance against macrolides (including erythromycin, roxithromycin, azithromycin, and clarithromycin). The unique detection of these genes on commercial chicken carcass samples, but not fecal samples, could indicate contamination from human, rinse bucket water, or other environmental sources during chicken processing activities (67, 68). For example, inadequate hygiene practices may result in contamination of chicken carcasses by human vendors while they handle the carcass during processing. Additionally, chicken carcasses may be exposed to contamination from surfaces or tools with which they are processed. Commercial carcass and rinse bucket water microbiomes and resistomes were not compositionally different from each other, indicating that rinse bucket water is likely a meaningful representation of pathogen and ARG exposures posed to consumers from the sale of commercial chicken carcasses at markets. Genes conferring resistance against aminoglycoside, folate pathway antagonist, macrolide, lincosamide, streptogramin B, quinolone, and tetracycline clinical antibiotic classes were detected among both carcass and rinse bucket samples. The presence of these ARGs in consumer poultry products may increase their risk of being mobilized by human microbes during food handling, preparation, or consumption.

This study had several strengths. The use of metagenomic sequencing to characterize samples at multiple points across the poultry value chain contributes novel insights and supports the use of an emerging "farm-to-fork approach" in foodborne disease research (63, 69, 70). Our study, nested within a larger study that mapped the poultry value chain prior to selecting sites for sampling and modelled transmission risk at various sites, helps to provide a comprehensive understanding of potential pathogen and ARG exposures within this system. This study also provides metagenomic characterization of poultry in Maputo City, Mozambique, a setting with limited poultry pathogen and ARG prevalence

data. Finally, distinguishing between household and commercial chickens contributes to understanding of microbiome and resistome differences between breeds and points along the poultry value chain. This study reflects the characteristically exploratory nature of "omics" approaches and provides foundational evidence for further investigation (71).

There were also some limitations to this study. First, relatively few samples were available from farms and markets, and no samples were available from other key points along the poultry value chain, including depots (where commercial chicks are purchased) and neighborhood food stalls, which limited the extent to which we could characterize hazards along the poultry value chain. Limited sample size overall may impact generalizability of these results for contexts beyond Maputo City, Mozambique, but our sampling of the multiple markets that exist in Maputo did provide a robust evaluation of poultry in this city. Local chicken and production site sample sizes were especially limited, which may result in lower sensitivity to detect key HR-ARGs and potentially pathogenic bacteria in those sample types. Samples across sites were not paired, so we are unable to conclude temporal shifts among individual chickens from farm to market and processing. This would be a valuable area for future investigations. Additionally, while untargeted sequencing offers the advantage of detecting a wide variety of strains and ARGs, metagenomic sequencing data does not represent exposure of consumers to pathogenic or antimicrobial-resistant bacteria due to the detection of non-infectious DNA fragments and low pathogen concentrations, as well as die-off between point of sampling and points of contact for consumers. As such, our findings represent hazards of foodborne illness that could be used to estimate risks of exposure and downstream sequalae (2). Metagenomic sequencing has low sensitivity for detection of rare pathogens and ARGs (72), especially compared with quantitative PCR; while we did not detect high risk beta-lactamase ARGs in our samples, these genes have been detected in poultry carcass samples in Mozambique (73).

Concentrations of genetic material in fecal and water samples are known to be highly variable, with much higher concentrations in fecal samples than water samples (74). This is consistent with our samples, in which sequencing depth was much greater among fecal samples than rinse bucket or carcass samples. Differences between carcass, rinse bucket, and fecal samples are therefore likely to be attributable at least in part to differences in limits of detection for different sample types rather than differences in actual contamination.

## Conclusion

Metagenomic analysis of chicken fecal and processing samples reveals distinct patterns of potential pathogen and ARG contamination along the poultry value chain in Maputo City, Mozambique. We found that potentially pathogenic bacteria and ARGs present in fecal samples are also present on carcasses sold to consumers, with distinct hazards potentially posed by household versus commercial chicken carcasses. Chicken carcasses are also contaminated with HR-ARGs that are not necessarily introduced from chicken feces. Our results indicate the potential for open-air/live markets as a critical juncture in the spread of pathogenic bacteria and ARGs in a community. Given that poultry production continues to increase in LMICs, culturally sensitive, market-based interventions that limit pathogen spread while maintaining these settings as affordable sources for fresh food are urgently needed.

## ACKNOWLEDGMENTS

We are grateful to the Maputo City Municipality for allowing us to conduct this study and to the study participants for their contributions. We would like to thank our in-country collaborators from the Center of Biotechnology at Eduardo Mondlane University, especially our enumerators, E. R. P. Bila, T. A. Cuinhane, D. H. Magaia, and M. M. Makongoro.

This work was supported by a Rollins School of Public Health (RSPH) Dean's Pilot and Innovation Award. Sample collection was supported by the Bill & Melinda Gates

Foundation (OPP 1189339). N.O. was supported by the National Institute of Environmental Health Sciences under Award Number 5T32ES012870. F.L. was supported by the National Institute of Allergy and Infectious Diseases of the National Institutes of Health under Award Number T32AI138952. K.J. was supported by the National Institute of Environmental Health Sciences under Award Numbers 5T32ES007032-37 and 5T32ES012870-15. The content is solely the responsibility of the authors and does not necessarily represent the official views of the National Institutes of Health.

## AUTHOR AFFILIATIONS

[1]Gangarosa Department of Environmental Health, Rollins School of Public Health, Emory University, Atlanta, Georgia, USA
[2]Veterinary Faculty, Universidade Eduardo Mondlane, Maputo, Mozambique
[3]Biotechnology Center, Universidade Eduardo Mondlane, Maputo, Mozambique
[4]Department of Environmental and Occupational Health Sciences, University of Washington, Seattle, Washington, USA

## AUTHOR ORCIDs

Natalie Olson  http://orcid.org/0000-0003-2175-4138
Frederica Lamar  http://orcid.org/0000-0003-3116-6727
Hermógenes Mucache  http://orcid.org/0000-0002-3110-8343
José Fafetine  http://orcid.org/0000-0002-9038-5922
Amélia Milisse  http://orcid.org/0000-0002-3546-5305
Denise R. A. Brito  http://orcid.org/0000-0001-8914-8631
Kelsey J. Jesser  http://orcid.org/0000-0002-4074-3170
Karen Levy  http://orcid.org/0000-0002-0968-9401
Matthew C. Freeman  http://orcid.org/0000-0002-1517-2572
Maya L. Nadimpalli  http://orcid.org/0000-0002-6526-116X

## FUNDING

| Funder | Grant(s) | Author(s) |
|---|---|---|
| Rollins School of Public Health Dean's Pilot and Innovation Award | | Maya L. Nadimpalli |
| Bill and Melinda Gates Foundation (GF) | OPP 1189339 | Karen Levy |
| | | Matthew Freeman |
| HHS \| NIH \| National Institute of Environmental Health Sciences (NIEHS) | 5T32ES007032-37, 5T32ES012870-15 | Kelsey J. Jesser |
| HHS \| NIH \| National Institute of Allergy and Infectious Diseases (NIAID) | T32AI138952 | Frederica Lamar |
| HHS \| NIH \| National Institute of Environmental Health Sciences (NIEHS) | 5T32ES012870 | Natalie Olson |

## DATA AVAILABILITY

All R code and data used for data analysis and visualization are available on GitHub (github.com/natalie0lson/cheep). Metagenome sequence data for all samples are available in the NCBI Sequence Read Archive (SRA) under number PRJNA1140288.

## ETHICS APPROVAL

The institutional review board at Emory University (IRB00108546) and the Research Council of the Veterinary Faculty at Eduardo Mondlane University determined that the ChEEP ChEEP study was exempt from human subjects review, and the Municipality of Maputo (Reference number 754/SG/426/GP/2019) authorized the ChEEP ChEEP study.

## ADDITIONAL FILES

The following material is available online.

### Supplemental Material

**Supplemental material (mSystems01037-24-s0001.docx).** Supplemental table and figures.

### Open Peer Review

**PEER REVIEW HISTORY (review-history.pdf).** An accounting of the reviewer comments and feedback.

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
