## [Reviewer comments · mSystems]

Farm-to-fork Changes in Poultry Microbiomes and Resistomes in Maputo City, Mozambique

Natalie Olson, Frederica Lamar, Hermógenes Mucache, José Fafetine, Joaquim Saíde, Amelia Milisse, Denise Brito, Kelsey Jesser, Karen Levy, Matthew Freeman, and Maya Nadimpalli

Corresponding Author(s): Maya Nadimpalli, Emory University School of Public Health

Review Timeline:

Submission Date:	July 31, 2024
Editorial Decision:	September 27, 2024
Revision Received:	November 22, 2024
Accepted:	December 2, 2024

Editor: Sarah Allard

Reviewer(s): Disclosure of reviewer identity is with reference to reviewer comments included in decision letter(s). The following individuals involved in review of your submission have agreed to reveal their identity: Pablo Tsukayama (Reviewer #1)

Transaction Report:

DOI: <https://doi.org/10.1128/msystems.01037-24>

Re: mSystems01037-24 (Farm-to-fork Changes in Poultry Microbiomes and Resistomes in Maputo City, Mozambique)

Dear Dr. Maya L Nadimpalli:

This is a well-done study from a region underrepresented in microbiome research. Although the sample size is small, the analysis is sound with appropriate caveats in place about the generalizability of the results. Please address the review comments, which highlight the need for more details in the methods section, additional discussion of potential contamination sources, and analysis and/or discussion of additional relevant ARGs. Please also double-check that potential pathogens are always discussed with the word "potential." I look forward to receiving the updated manuscript.

Revision Guidelines

Sincerely,
Sarah Allard
Editor
mSystems

Reviewer #1 (Comments for the Author):

This study uses metagenomic (a.k.a. shotgun) sequencing to investigate the microbiomes and resistomes present in poultry along the supply chain in Maputo City, Mozambique. It highlights the differences in bacterial communities and antimicrobial resistance genes (ARGs) between commercial and local chickens, and tracks the persistence of pathogens and ARGs from production sites to consumer-end carcasses. The study also identifies open-air markets as critical points for exposure to high-risk ARGs, suggesting potential sources of contamination beyond the poultry itself, such as rinse bucket water.

Major Strengths:

1. **Relevance and Impact:** The manuscript addresses an important public health issue in a low- and middle-income country (LMIC) setting, focusing on antimicrobial resistance (AMR), a critical global health challenge. The use of a novel "farm-to-fork" approach in this context is unique and provides valuable insights into pathogen and ARG transmission risks, sparking curiosity and engagement.
2. The authors use metagenomic sequencing to provide a comprehensive overview of bacterial communities and resistomes across different stages of the poultry supply chain.
3. The study highlights the unique challenges of poultry production in Mozambique, including poor hygiene practices at open-air markets, which is relevant for public health interventions in similar settings.

Major Concerns:

1. **Sample size:** Although the study provides valuable findings, the limited number of samples, especially from local chickens at markets and processing stages, could reduce the generalizability of the results. The authors should clarify the representativeness of the sampling design and address the potential limitations of small sample sizes.
2. **Lack of paired sampling:** The lack of paired samples limits the ability to track changes in microbiomes and resistomes from farm to market in the same chickens. This reduces the strength of the conclusions regarding temporal changes along the supply chain. Adding a discussion on the limitations this poses for interpreting results would strengthen the manuscript.
3. **Lack of details on DNA extraction protocols:** Metagenomic-based studies are particularly susceptible to differences in DNA extraction protocols, which need to be listed in the main text. Please include details on how samples were preserved at the point of collection, how they were stored, which DNA extraction method was used, and whether any filtering/concentration steps were performed on the water samples. Also, did they measure DNA concentration after extraction? Which kits were used to prepare the Illumina DNA libraries?
4. **Lack of details on bioinformatic analyses:** Software tools are listed, but to ensure reproducibility of the results, authors should also include the version used, as well as the main parameters used to run their analyses.
5. **Source of contamination:** While the study points to rinse bucket water as a potential source of ARG contamination, there is limited direct evidence to support this claim. The authors could benefit from discussing alternative contamination sources, such as market environments and vendor practices, more thoroughly.
6. There are no statements regarding the approval of research involving animals.

Reviewer #2 (Comments for the Author):

This study compares the pathobiome and resistome of commercial vs. local poultry from the production site to the point of sale in urban Mozambique. The results from these metagenomic samples can be used to target public health and food safety interventions to maximally reduce consumers' pathogen exposures. This is a well written manuscript with interesting and useful data visualizations. However, there are a few major points of concern, the first being the different sample sizes for commercial vs. local samples along the supply chain. Is the smaller sample size for local chickens, especially for carcass samples, driving some of the differences in pathogen and ARG abundance? Second, though previous analyses from the parent study have been published, the description of the study design and approach is lacking from the current manuscript. Third, I'm surprised at the omission of beta-lactamase and other ARGs that confer resistance to last-line drugs from the list of high-risk ARGs in this analysis. The authors might reconsider adding ESBL genes (blaCTX, blaTET, blaOXA, blaNDM) and a colistin gene (mcr-1) to the list of HR-ARGs analyzed (or add this to SI). See other specific comments below.

Methods

Line 118: At what level were samples pooled? At each location? By type of poultry? Were there multiple types of chickens (broiler/layer/local) at a single location?

Lines 117-126: Several details about the study approach (e.g. site selection for farms/markets/households, how/when samples were pooled, how samples were processed, DNA extraction method used) are missing. Even if previously described in other

studies, it would be helpful to have a brief description of these details in addition to citing the parent study.

Line 127: I'd suggest removing the subheader "Experimental Design" as that doesn't quite fit for this paragraph.

Lines 128-131: Can the authors add a table in SI with metadata about the samples that didn't have sufficient DNA for this analysis? Were these samples systematically different from the 101 samples included in this analysis?

Line 181: Specifically, what is defined as the "referent group" and the "comparator" in this analysis?

I think a table with metadata about the farms, markets, and households included in this analysis should be added, if available.

How many poultry were at each farm? How many stands were at each market? How many people and chickens per household?

Any other animals owned at the households?

Results

Line 208: Please report the exact p-value instead of $p < 0.05$

Lines 218-219: Sentence beginning with "Commercial and local chickens are sold at the same markets..." would fit best in the methods section, along with other details about context.

Figure 1: I might suggest a different icon for the processing/carcass part of the schematic - I think this image is too representative of the point of consumption rather than the point of purchase. You could do a farm icon for "Farm", a market icon for "Market", then the chicken icon for "carcass"?

Figures 2 & 3 caption: In panel B, the statistical approach used to estimate differential abundance and the specific estimate represented in the figure should be described more clearly in the caption.

Figures 2 & 3: In panel B, x axis label should be added (relative abundance)?

Figure 3: In panel C, y axis text should be spaced out so it's more legible and an x axis label should be added (I believe each label represents a unique pooled sample, but that isn't clear based on the caption or labels in the figure itself).

Figure 4 caption: Generally, I think the approach of describing a key result in the figure caption is not as helpful as a specific description of what each plot represents, especially for the panel descriptions. The panel captions should describe what the plot is, not provide an interpretation of the results visualized in the plot. Also FPGE should be defined in the caption for panel A.

Figure 4: All text size (axis, facet and legend labels) should be increased, it's difficult to see. In panel A, facet labels are repeated above and below the plot. Also, is panel A missing the category "Local/Market/Carcass"? If so, can the authors clarify why this is?

Discussion

Lines 298-299: I don't think this study actually characterized exposures at each point in the supply chain - only at farms and markets. The authors later mention this as a limitation, but it should be clarified here. Maybe change "at each point in the supply chain" to "at multiple points in the supply chain".

Lines 347-350: why were beta-lactamase genes excluded from the list of HR-ARGs?

Comment	Response
E1. Please address the review comments, which highlight the need for more details in the methods section, additional discussion of potential contamination sources, and analysis and/or discussion of additional relevant ARGs. Please also double-check that potential pathogens are always discussed with the word "potential." I look forward to receiving the updated manuscript.	In response to reviewers' comments, we have added additional discussion of potential contamination sources and addressed additional relevant ARGs which were not detected in our samples. We confirm that pathogens are consistently referred to as "potential pathogens" throughout the text. Locations of these specific in-line edits are provided in the responses to individual reviewers below. Throughout the response to reviewers, new text is indicated with underlining.
R1.1. Sample size: Although the study provides valuable findings, the limited number of samples, especially from local chickens at markets and processing stages, could reduce the generalizability of the results. The authors should clarify the representativeness of the sampling design and address the potential limitations of small sample sizes.	We have addressed potential limitations of small sample sizes by adding text to our discussion on lines 410-418: "First, relatively few samples were available from farms and markets and no samples were available from other key points along the poultry value chain, including depots (where commercial chicks are purchased) and neighborhood food stalls, which limited the extent to which we could characterize hazards along the poultry value chain. Limited sample size overall may impact generalizability of these results for contexts beyond Maputo City, Mozambique, but our sampling of the multiple markets that exist in Maputo did provide a robust evaluation of poultry in this city. Local chicken and production site sample sizes were especially limited which may result in lower sensitivity to detect key HR-ARGs and potentially pathogenic bacteria in those sample types."
R1.2. Lack of paired sampling: The lack of paired samples limits the ability to track changes in microbiomes and resistomes from farm to market in the same chickens. This reduces the strength of the conclusions regarding temporal changes along the supply chain. Adding a discussion on the limitations this poses for interpreting results would strengthen the manuscript.	We have now discussed limitations posed by lack of paired sampling for interpretation of results by adding the following text to lines 418- 420: "Samples across sites were not paired, so we are unable to conclude temporal shifts among individual chickens from farm to market and processing. This would be a valuable area for future investigations."
R1.3. Lack of details on DNA extraction protocols: Metagenomic-based studies are particularly susceptible to differences in DNA extraction protocols, which need to be listed in the main text. Please include details on how samples were preserved at the point of collection, how they were stored, which DNA extraction method was used, and whether any filtering/concentration steps were performed on the water samples. Also, did they measure DNA concentration after extraction? Which kits were used to prepare the Illumina DNA libraries?	We have added substantial detail on how samples were preserved at point of collection and stored, DNA extraction method, filtering/concentration steps for water samples, and DNA concentration measurement threshold on lines 121-146: "Sample collection and processing methods are described in detail in prior ChEEP ChEEP study publications^{23,24}. Sterile spatulas were used to collect pooled fecal samples from broiler and layer housing at small-scale farms, yards and holding cages at households, and shared holding cages at markets; samples and a daily field blank were transported to the laboratory on ice and processed the same day. For each chicken breed and site, four fecal samples were pooled by equal weights. DNA was extracted from ~ 1g of each pooled fecal sample using Qiagen PowerBead DNA extraction tubes²³."

	Rinses of chicken carcasses, hereafter referred to as “carcass samples,” (n=75) were collected from broiler, layer, and local chickens at markets between 2019 and 2021²³. Raw carcasses were placed in Whirl-Pak poultry rinse bags at the market, transported to the laboratory on ice, and processed the same day. Individual carcasses were rinsed with 400 mL 0.1% buffered peptone water in stomacher bags by shaking and moving in an arc motion for 1 minute and 100 mL of the resulting solution was aliquoted into two 50-mL conical tubes and centrifuged. Resultant pellets were resuspended in 1 mL 1X phosphate buffered saline (PBS), then 125 µL of each aliquot (250 µL total for each carcass) were transferred to PowerBead DNA extraction tubes. Water collected from rinse buckets used by vendors for processing broiler chickens was additionally collected from three markets between 2021 and 2022²⁴; hereafter referred to as “rinse bucket samples.” We refer to carcass and rinse bucket samples collectively as “processing samples.” To preserve the integrity of genetic material in our samples, Qiagen DNEasy PowerSoil Kit extraction buffer “Solution C1” was added to each sample. Six freeze-thaw cycles were completed to break open Cryptosporidium spp. oocysts. DNA was extracted from all samples using DNEasy PowerSoil Kit and stored at -80°C. Invitrogen Qubit fluorometry and NanoDrop were used to measure the DNA concentrations in each extract, and all samples used for sequencing were >10 ng/µL. We have included Illumina DNA library preparation kit information on line 154: “The NEBNext® Ultra™ II DNA Library Prep Kit was used for library preparation.”
R1.4. Lack of details on bioinformatic analyses: Software tools are listed, but to ensure reproducibility of the results, authors should also include the version used, as well as the main parameters used to run their analyses.	We have added details on versions and parameters used for bioinformatic analyses on lines 157-181: We removed adapters and low-quality reads from raw read files using BBDuk version 38.9²⁶. We used Bowtie2 version 2.5.1²⁷ with NIH Genome Reference Consortium reference genomes to align and remove host and eukaryotic contaminant DNA sequences [...]. We used a reduced 500GB NCBI RefSeq taxonomic database³⁶ to conduct taxonomic profiling using Kraken2 version 2.1.2 with default k-mer length of 35³⁷ and used Bracken version 2.7.0 (Bayesian Re-estimation of Abundance after Classification with Kraken)³⁸ using default settings to estimate relative abundance of classified bacteria. [...] We used k-mer alignment (KMA) version 1.4.9⁴¹ to align trimmed and host-decontaminated reads to the ResFinder database (downloaded June 2, 2023)⁴² using a threshold of 95% identity and 90% coverage⁴¹ [...] To estimate the total number of unique organisms present in our data, we estimated genome equivalents for each sample metagenome using Microbe Census version 1.1.1⁴⁴.
R1.5. Source of contamination: While the study points to rinse bucket water as a potential source of ARG contamination, there is limited direct evidence to support this claim.	We have now included more thorough discussion of potential alternative contamination sources such as market environment and vendor practices on lines 388-391:

The authors could benefit from discussing alternative contamination sources, such as market environments and vendor practices, more thoroughly.	“For example, inadequate hygiene practices may result in contamination of chicken carcasses by human vendors while they handle the carcass during processing. Additionally, chicken carcasses may be exposed to contamination from surfaces or tools with which they are processed.” This is also addressed on lines 329-334: “Rinse bucket water is a likely source of additional contamination, especially given that 100% of chickens tested positive via PCR for Campylobacter coli/jejuni and E. coli after being rinsed, despite fewer chickens testing positive for these species prior to rinsing²⁴. Additionally, other environmental sources such as vendors, surfaces, and pests, could explain the greater diversity of potential pathogens and ARGs observed on carcasses.”
R1.6. 6. There are no statements regarding the approval of research involving animals.	No IACUC approvals were required for this study because all sampling was from chickens that had already been slaughtered for sale. Information about other ethical review and approvals have been included on lines 468 – 471: “The institutional review board at Emory University (IRB00108546) and the Research Council of the Veterinary Faculty at Eduardo Mondlane University determined that the ChEEP ChEEP study was exempt from human subjects review, and the Municipality of Maputo (Reference number 754/SG/426/GP/2019) authorized the ChEEP ChEEP study.”
R2.1. However, there are a few major points of concern, the first being the different sample sizes for commercial vs. local samples along the supply chain. Is the smaller sample size for local chickens, especially for carcass samples, driving some of the differences in pathogen and ARG abundance?	The revised manuscript includes more thorough discussion of limitations due to sample size [See response to R1.1]
R2.2. Second, though previous analyses from the parent study have been published, the description of the study design and approach is lacking from the current manuscript.	We have included more thorough details of sampling design and collection methods [See response to R1.3]
R2.3. Third, I'm surprised at the omission of beta-lactamase and other ARGs that confer resistance to last-line drugs from the list of high-risk ARGs in this analysis. The authors might reconsider adding ESBL genes (blaCTX, blaTET, blaOXA, blaNDM) and a colistin gene (mcr-1) to the list of HR-ARGs analyzed (or add this to SI).	Beta-lactamase and other important ARGs were considered, but were not detected in our samples (See all HR-ARGs we screened for listed in SI Table 3) We also addressed low sensitivity to detect these ARGs on lines 426- 429: “Metagenomic sequencing has low sensitivity for detection of rare pathogens and ARGs⁷³, especially compared with quantitative PCR; while we did not detect high risk beta-lactamase ARGs in our samples, these genes have been detected in poultry carcass samples in Mozambique⁷⁴.”
R2.4. Line 118: At what level were samples pooled? At each location? By type of poultry? Were there multiple types of chickens (broiler/layer/local) at a single location?	We have included more thorough description of sample collection and pooling methods on lines 119-128. This section now reads as follows: “Pooled fecal samples (n=136) were collected from broiler, layer, and local chickens at farms, households, and markets. Commercial and local chickens are sold at the same markets

	in Maputo City; fecal samples were collected from cages after arriving at markets. Sample collection and processing methods are described in detail in prior ChEEP ChEEP study publications^{23, 24}. Sterile spatulas were used to collect pooled fecal samples from broiler and layer housing at small-scale farms, yards and holding cages at households, and shared holding cages at markets; samples and a daily field blank were transported to the laboratory on ice and processed the same day. For each chicken breed and site, four fecal samples were pooled by equal weights. DNA was extracted from ~ 1g of each pooled fecal sample using Qiagen PowerBead DNA extraction tubes.
R2.5. Lines 117-126: Several details about the study approach (e.g. site selection for farms/markets/households, how/when samples were pooled, how samples were processed, DNA extraction method used) are missing. Even if previously described in other studies, it would be helpful to have a brief description of these details in addition to citing the parent study.	We have included more thorough details of sample collection and DNA extraction methods [See response to R1.3]
R2.6. Line 127: I'd suggest removing the subheader "Experimental Design" as that doesn't quite fit for this paragraph.	Sub-header "Experimental Design" removed from line 118.
R2.7. Lines 128-131: Can the authors add a table in SI with metadata about the samples that didn't have sufficient DNA for this analysis? Were these samples systematically different from the 101 samples included in this analysis?	Metadata for samples with insufficient DNA for analysis are now included in SI Table 2 and referenced in lines 147-148: "Extracted DNA from a subset of these samples (N=101) was available in sufficient quantity (>10 ng/μL threshold concentration) for metagenomic sequencing (SI Table 2)."
R2.8. Line 181: Specifically, what is defined as the "referent group" and the "comparator" in this analysis?	Reference and comparator groups are defined for each comparison in the "Results" section, for example on lines 256-260: "Commercial chickens sampled at markets harbored a greater number of bacterial species ($p=0.002$) and a greater number of potentially pathogenic species ($p=0.006$) than commercial chickens sampled at the farm level, though we observed no differences in species richness, diversity, or taxa abundance among local chicken feces sampled at households versus at markets. "
R2.9. I think a table with metadata about the farms, markets, and households included in this analysis should be added, if available. How many poultry were at each farm? How many stands were at each market? How many people and chickens per household? Any other animals owned at the households?	Unfortunately, this information is not available.
R2.10. Line 208: Please report the exact p-value instead of $p<0.05$	Exact p-values are now reported on lines 260-263: "Potentially pathogenic Listeria spp. ($P_{adj}=0.035$), Salmonella spp. ($P_{adj}<0.001$), Klebsiella spp. ($P_{adj}<0.001$), Enterobacter spp. ($P_{adj}<0.001$), Pseudomonas spp. ($P_{adj}<0.001$), Vibrio spp. ($P_{adj} = <0.001$), and Treponema spp. ($P_{adj}<0.001$) were relatively less abundant among commercial chickens sampled at markets versus at farms after adjusting for FDR."
R2.11. Lines 218-219: Sentence beginning with "Commercial and local chickens are sold	We moved this sentence to the methods section on lines 120-121:

at the same markets..." would fit best in the methods section, along with other details about context.	"Commercial and local chickens are sold at the same markets in Maputo City; fecal samples were collected from cages after arriving at markets."
R2.12. Figure 1: I might suggest a different icon for the processing/carcass part of the schematic - I think this image is too representative of the point of consumption rather than the point of purchase. You could do a farm icon for "Farm", a market icon for "Market", then the chicken icon for "carcass"?	We appreciate this suggestion, but we feel that the chicken icon might be misleading for carcass samples as it depicts a live chicken with feathers etc. Carcass samples were fully processed chicken meat purchased by consumers (slaughtered, feathers removed, etc.). We have added icons depicting sites at which samples were collected (farm, household, or market) to clarify that carcass samples were collected from chicken carcasses at the point of purchase at the market.
R2.13. Figures 2 & 3 caption: In panel B, the statistical approach used to estimate differential abundance and the specific estimate represented in the figure should be described more clearly in the caption.	We have updated the figure captions to describe the statistical approach for estimation of differential abundance and interpretation of estimates: "Figure 2. Commercial and local chicken fecal microbiomes & resistomes are compositionally distinct at production sites. A) Non-metric multi-dimensional scaling (NMDS) plot of Bray Curtis distances for pathogen composition. B) Differences in relative abundance of pathogens among local versus commercial (reference) chickens, modeled using beta-binomial regression. C) Heat map of high-risk antimicrobial resistance gene (HR-ARG) abundances among commercial and local chickens."
R2.14. Figures 2 & 3: In panel B, x axis label should be added (relative abundance)?	Thank you for noting this omission. We have added the x axis label for relative abundance.
R2.15. Figure 3: In panel C, y axis text should be spaced out so it's more legible and an x axis label should be added (I believe each label represents a unique pooled sample, but that isn't clear based on the caption or labels in the figure itself).	We appreciate these suggestions. We have spaced out the y-axis text and included x-axis label "Sample".
R2.16. Figure 4 caption: Generally, I think the approach of describing a key result in the figure caption is not as helpful as a specific description of what each plot represents, especially for the panel descriptions. The panel captions should describe what the plot is, not provide an interpretation of the results visualized in the plot. Also FPGE should be defined in the caption for panel A.	Added description of plot and FPGE defined in caption: "Figure 4. Consumers are exposed to clinically important high-risk antimicrobial resistance genes (HR-ARGs) via chicken carcasses that are not necessarily introduced from chicken feces. A) Violin plot depicting HR-ARG abundance measured in fragments per genome equivalent (FPGE) of local and commercial chicken samples at production, market, and processing. HR-ARGs are abundant at multiple points in the supply chain, especially among commercial chickens. Note that local carcass samples are missing from this plot as no HR-ARGs were detected in these samples. B) Heat map of HR-ARG abundances among commercial fecal and carcass samples. HR-ARGs ant(2)-Ia, aph(3)-IIIa, and mphE were identified in commercial carcass but not market fecal samples."
R2.17. Figure 4: All text size (axis, facet and legend labels) should be increased, it's difficult to see. In panel A, facet labels are repeated above and below the plot. Also, is panel A missing the category "Local/Market/Carcass"? If so, can the authors clarify why this is?	We have increased text sizes and added clarification of missing "local/market/carcass" category in the plot caption: "Note that local carcass samples are missing from this plot as no HR-ARGs were detected in these samples."

R2.18. Lines 298-299: I don't think this study actually characterized exposures at each point in the supply chain - only at farms and markets. The authors later mention this as a limitation, but it should be clarified here. Maybe change "at each point in the supply chain" to "at multiple points in the supply chain".	We clarified points of exposure by changing "each point in the supply chain" to "multiple points in the supply chain" on lines 318-319: "We employed metagenomic shotgun sequencing to characterize and compare poultry microbiomes and resistomes at multiple points in the supply chain in Maputo City, Mozambique."
R2.19. Lines 347-350: why were beta-lactamase genes excluded from the list of HR-ARGs?	Beta-lactamase genes were included in our HR-ARG list (see new table included in supplemental materials) but were not detected in our samples. We have now addressed this point in the revised manuscript [See response to R2.3]

Re: mSystems01037-24R1 (Farm-to-fork Changes in Poultry Microbiomes and Resistomes in Maputo City, Mozambique)

Dear Dr. Maya L Nadimpalli:

Your manuscript has been accepted, and I am forwarding it to the ASM production staff for publication. Your paper will first be checked to make sure all elements meet the technical requirements. ASM staff will contact you if anything needs to be revised before copyediting and production can begin. Otherwise, you will be notified when your proofs are ready to be viewed.

Sincerely,
Sarah Allard
Editor
mSystems